# Effects of Diet and Lifestyle on Audio-Vestibular Dysfunction in the Elderly: A Literature Review

**DOI:** 10.3390/nu14224720

**Published:** 2022-11-08

**Authors:** Hsin-Lin Chen, Ching-Ting Tan, Chen-Chi Wu, Tien-Chen Liu

**Affiliations:** 1Department of Surgical Oncology, National Taiwan University Cancer Center Hospital, Taipei 100, Taiwan; 2Department of Otolaryngology, National Taiwan University Hospital, Taipei 100, Taiwan; 3Department of Medical Research, National Taiwan University Hospital Hsin-Chu Branch, Hsinchu 302, Taiwan; 4Department of Medical Genetics, National Taiwan University Hospital, Taipei 100, Taiwan; 5Graduate Institute of Clinical Medicine, National Taiwan University College of Medicine, Taipei 100, Taiwan

**Keywords:** audio-vestibular dysfunction, age-related hearing loss, tinnitus, vertigo, diet, nutritional intervention, lifestyle modification

## Abstract

Background: The world’s age-related health concerns continue to rise. Audio-vestibular disorders, such as hearing loss, tinnitus, and vertigo, are common complaints in the elderly and are associated with social and public health burdens. Various preventative measures can ease their impact, including healthy food consumption, nutritional supplementation, and lifestyle modification. We aim to provide a comprehensive summary of current possible strategies for preventing the age-related audio-vestibular dysfunction. Methods: A PubMed, Embase, and Cochrane review databases search was conducted to identify the relationship between diet, lifestyle, and audio-vestibular dysfunction. “Diet”, “nutritional supplement”, “lifestyle”, “exercise”, “physical activity”, “tinnitus”, “vertigo” and “age-related hearing loss” were used as keywords. Results: Audio-vestibular dysfunction develops and progresses as a result of age-related inflammation and oxidative stress. Diets with anti-inflammatory and antioxidant effects have been proposed to alleviate this illness. A high-fat diet may induce oxidative stress and low protein intake is associated with hearing discomfort in the elderly. Increased carbohydrate and sugar intake positively correlate with the incidence of audio-vestibular dysfunction, whereas a Mediterranean-style diet can protect against the disease. Antioxidants in the form of vitamins A, C, and E; physical activity; good sleep quality; smoking cessation; moderate alcohol consumption; and avoiding noise exposure are also beneficial. Conclusions: Adequate diet or nutritional interventions with lifestyle modification may protect against developing audio-vestibular dysfunction in elderly individuals.

## 1. Introduction

The elderly population is the fastest growing population globally. The World Health Organization estimated that the proportion of people over the age of 60 will nearly double between 2015 and 2050, and by 2030, 20% of the population will be 65 years of age or older (1). This trend is of medical concern as aging causes a decline in multiple organ system functions, such as hearing loss, refractive errors, depression, dementia, and multiple chronic disorders [1]. Thus, increased life expectancy will be accompanied by “expansion of morbidity” [2,3].

Age-related hearing loss (ARHL) and age-related vestibular loss (ARVL) are common conditions that have deleterious consequences on patients’ health and quality of life. ARHL or presbycusis usually affects both ears gradually and equally and is frequently reported in the elderly, with a prevalence ranging from 30% of individuals aged 65–74 years to more than 40–60% of people aged 75 years or older [4,5]. ARHL cases and disability-adjusted life years (DALYs) worldwide nearly doubled between 1990 and 2019 [6]. Additionally, males are more susceptible to the disease than women. Modifiable risk factors that can increase the likelihood of ARHL include noise exposure, smoking, and medical comorbidities such as hypertension, diabetes, and cerebrovascular and cardiovascular disease [7]. 

Dizziness and imbalance are major complaints among the elderly population, and peripheral vestibular dysfunction is one of their most common causes [8]. Age-related vestibular loss, also known as presbyastasis or presbyequilibrium, commonly occurs during the normal aging process [9,10]; however, in contrast to ARHL, few studies have reported its prevalence. A population-based study in the United States reported that 24% of people over the age of 72 years experienced dizziness [11].

ARHL and ARVL lead to significant health burdens and reduced quality of life [12]. Dizziness and imbalance are associated with an elevated risk of accidental falls and injuries. According to the National Institute of Deafness and Other Communication Disorders of the National Institutes of Health, falls account for over 50% of all accidental deaths among the elderly [13]. 

Prevention and treatment strategies for these diseases are limited; however, patient nutritional status has been identified as a modifiable risk factor. A previous article ever depicted the possible association between nutritional status and ARHL. However, the article did not mention ARVL and the association of lifestyle and ARHL and ARVL [14]. Therefore, we aimed to provide a review and summary of current possible strategies for preventing the adverse effects associated with ARHL and ARVL through nutritional and lifestyle modification.

## 2. Materials and Methods

### 2.1. Search Strategy

Two authors (Chen HL and Tan CT) independently searched PubMed, Embase, and Cochrane review databases for relevant articles published from database inception to December 2021. The database search was conducted to identify the relationship between diet, lifestyle, and audio-vestibular dysfunction. “Diet,” “nutritional supplement,” “lifestyle,” “exercise,” “physical activity,” “tinnitus,” “dizziness,” “vertigo,” “age-related hearing loss,” “age-related vestibular loss,” and “aging” were included as keywords. We imposed no language or temporal restrictions on any of the searches. The databases were searched by 2 authors (Chen HL and Tan CT) independently and were reviewed by the 2 corresponding authors (Wu CC and Liu TC). 

### 2.2. Quality Assessments

The quality of the included studies was measured using the Newcastle–Ottawa Scale (NOS) [15]. The NOS score ranged from 0 (lowest quality) to 9 (highest quality). The NOS score was assessed by 2 authors independently (Chen HL and Wu CC), and the disagreements were resolved through a consensus.

## 3. Results

### 3.1. Literature Search

Figure 1 illustrates the flowchart of the literature search [16]. The initial database search yielded 1664 studies. After duplicates were removed, 1572 studies were identified and further screened. Finally, there was a full text of 66 studies and then 42 studies were removed with reasons.

### 3.2. Quality Assessment

The quality assessment of the included studies was conducted using the NOS, with scores ranging from 0 (lowest) to 9 (highest) points. Appendix A list the detailed NOS results. The NOS score of included studies relating to the effects of diets to ARHL, ARVL, and related components was 5 in 3 studies, 6 in 5 studies, 7 in 3 studies, and 8 in 3 studies. Meanwhile, the NOS score of included studies related to the effects of lifestyles to ARHL, ARVL, and related components was 4 in 6 studies, 5 in 5 studies, 6 in 2 studies.

First of all, we make a simple review of pathophysiology and etiology of age-related hearing and vestibular loss, which may be related to inflammation and oxidative stress causing by the aging process. ARHL and ARVL development are also largely influenced by genetic and environmental factors such as eating habits and lifestyle. 

### 3.3. Pathophysiology and Etiology of Age-Related Hearing and Vestibular Loss

ARHL etiology has been extensively studied; however, little is known about the mechanism of ARVL. The auditory and vestibular systems rely on the same sensory organs in the inner ear. The inner ear comprises the cochlea, which is responsible for hearing, and the vestibular end organs, which are responsible for balance. The inner ear is also susceptible to aging effects. Primary age-related cochlear changes include decreased hair cell and neuron levels in the cochlear nuclei and auditory center of the brain. In contrast, changes in the peripheral vestibular system include loss of neurons and hair cells in the otolith organs and semicircular canals [13,17].

Insufficient cochlear blood flow is thought to be responsible for hair cell damage and hearing loss in the elderly. Free radical formation, which results from increased oxidative stress, and inflammation typically occur with age and can lead to impaired vascular function, as well as cochlear and vestibular damage [18]. Although the terms ARHL and ARVL are used to imply age-related audio-vestibular dysfunction, multiple etiologies of this disease exist.

#### 3.3.1. Inflammation

Inflammation is a hallmark of aging. The term “inflammaging” is defined as chronic low-grade inflammation that worsens with age and contributes to various age-related pathologies [19]. Inflammation is an indicator of accelerated aging and arises from immune or cellular senescence. Consequently, inflammation is also associated with ARHL [20,21]. 

Immune surveillance occurs in the cochlea [22,23] and stimulates circulating immunocyte recruitment [24,25]. Immunocytes enter the cochlea through post-capillary venules in the spiral ligament, causing cytokine and chemokine infiltration and expression in the spiral ligament, which may be related to age-related degenerative changes in these structures [26,27,28].

However, the exact role of inflammation in ARVL has not yet been investigated; however, inflammatory molecular mediators are present in the vestibular end organs [17]. To date, no effective medical treatment for ARVL exists; however, emerging treatments such as mitochondrial antioxidants or caloric restriction, which have shown promise in preventing ARHL, may also play a role in treating ARVL [29,30,31].

#### 3.3.2. Oxidative Stress

Oxidative damage results in age-related damage by forming free radicals and reactive oxygen species (ROS) [32]. ROS and free radicals damage DNA, break down lipid and protein molecules, and trigger cell death, which results in cochlear, especially hair cell, damage [33].

Growing evidence suggests that the effect of oxidative stress can induce macromolecule damage, such as mitochondrial DNA (mtDNA) mutations. mtDNA accumulates mutations/deletions which impairs mitochondrial function and induces cochlear cell apoptosis and ARHL [34].

Several studies have implicated the role of oxidative stress in initiating and progressing inner ear damage in animal models [17]. However, few studies have examined oxidative stress effects in progressing ARHL and ARVL simultaneously. Current reports suggest that the cochlea and vestibular end organs utilize overlapping but with distinct antioxidant enzymes that are affected by aging [7]. Although different antioxidant systems may be involved, these results indicate that oxidative stress likely plays a role in both ARHL and ARVL [17].

#### 3.3.3. Other Factors: Genetics, Environment, and Medication

Genetic studies have identified several genes that contributed to ARHL and ARVL pathogenesis, including those related to antioxidant defense and atherosclerosis [35,36]. Twin and family studies have revealed that heritable risk factors increase the risk of ARHL by 25% to 75%. Recent reports that incorporated genome-wide association studies (GWAS), gene set enrichment analyses, transcriptomic and epigenomic data from the mouse cochlea, and immunohistochemistry in the mouse cochlea have implicated the role of certain genes in cochlear metabolic, sensory, and neuronal functioning [37,38]. Candidate genes associated with ARHL development were identified by GWAS and include *ILDR1*, *ZNF318*, *NID2*, and *ARHGEF28* [37,39,40]. An increasing number of associated genes continue to be discovered. One meta-analysis study identified 48 associated loci, including 10 novel associations and 8 missense SNPs related to hearing loss, highlighting the importance of the stria vascularis in the mechanism of hearing impairment [41].

In contrast, no definite genes are directly linked to ARVL; however, genetic predisposition to antioxidant defense and atherosclerosis may play an important role in age-related vestibular end-organ dysfunction, due to oxidative stress and mutations in mtDNA [8,42,43]. Further studies are required to identify additional genes related to ARHL and ARVL.

In addition, other extrinsic modifiable factors, such as diet [44] and exercise [45], may influence ARHL and ARVL. Exposure to ototoxins [46] and noise [47] may also exacerbate acquired hearing loss and vestibular dysfunction, and their synergistic effects have a greater effect on ARHL [48]. Figure 2 summarizes the pathogenetic mechanisms of age-related hearing and vestibular loss. Aging contributes to inflammation and accumulation of oxidative stress, which results in changes in proteostasis, cellular senescence, impaired autophagy, and mitochondrial dysfunction. Ultimately, the inner ear is damaged by detrimental substances.

### 3.4. The Effects of Diet on the Audio-Vestibular Dysfunction of the Elderly

Aging induces inflammation and oxidative stress and affects microcirculation throughout the body. Chronic inflammation is observed during aging and has been linked to ARHL and ARVL severity. Furthermore, ARHL and ARVL development has also been largely influenced by genetic and environmental factors such as eating habits and lifestyle. Diet has been documented to heavily influence hearing and balance status since it can mediate age-related changes in the inner ear [49] by modulating systemic inflammation. Previous studies have linked pro-inflammatory food groups to ARHL; however, none have evaluated the anti-inflammatory effects of a balanced diet on ARHL attenuation [50]. Several emerging dietary patterns have been proposed to promote health; however, the beneficial effects of these diets on age-related audio-vestibular dysfunction need to be further explored.

#### 3.4.1. Low Fat and Low Cholesterol Diet

Several studies have revealed the harmful effects of a high-fat diet on the inner ear, wherein it induces oxidative stress, mitochondrial damage, and apoptosis [51]. This highlights a link between high-fat diets and the increased risk of developing ARHL [51,52]. An ARHL animal model reported an increased incidence of hearing loss in animals fed a high-fat diet, and the incidence significantly increased when accompanied by intermittent hypoxic conditions or d-galactose injection [53]. Thus, animal models can be used to develop preventive and treatment strategies for ARHL [54]. 

Moreover, cholesterol-rich diets have been shown to affect hearing loss [52]. Elevated serum lipid levels were observed in patients presenting with sudden sensory hearing loss and tinnitus [55,56,57]; however, this association has not been confirmed in patients with ARHL. Previous studies have suggested a relationship between a diet rich in saturated fats and poor hearing [58,59], and high saturated fat intake was linked to hearing loss, possibly via cardiovascular disease pathways [60,61,62,63].

Adiponectin is the most abundant adipose tissue-derived cytokine in the body, and low levels reflect obesity-induced adipose dysfunction [64]. Adiponectin was also implicated in several physiological processes and reported to exert anti-inflammatory [65] and anti-apoptotic effects [66]. Notably, obesity and other comorbidities contribute to ARHL by decreasing adiponectin levels. Our previous clinical cohort study highlighted the protective role of adiponectin against ARHL development [67]. The interaction between adiponectin and its type 1 receptor in the cochlea may exhibit protective effects against hair cell apoptosis [68,69]. Low-fat diets, wherein 30% or less of the calories are derived from fat, may also protect against ARHL. 

The four main subtypes of dietary fats are as follows: polyunsaturated, mono-unsaturated, trans, and saturated. Mono-unsaturated fatty acids (MUFA) and polyunsaturated fatty acids (PUFA) are considered beneficial, whereas saturated and trans fats are considered unhealthy since they increase low-density lipoprotein (LDL) levels. Also, a low-saturated fat diet may protect against hearing loss [70], wherein high polyunsaturated fat intake decreases the incidence of subjective and audiometric hearing loss [63]. An inverse relationship between PUFA and fish consumption and the incidence of hearing loss has also been reported [71]. Consuming fish rich in omega-3 fatty acids was found to reduce presbycusis frequency by 42% after a 5-year follow up. Long-chain n-3 PUFAs, which are abundant in fish, also lower the incidence of hearing loss [72,73].

#### 3.4.2. High Protein Diet

Insufficient protein intake results in ototoxic side effects [74]. Previous studies have suggested that various peptides, such as insulin, orexin, and leptin, play a critical role in regulating hippocampal synaptic plasticity and enhancing cognitive function [75]. Thus, low protein intake may impair neural function in the auditory system [74]. Kim et al. reported an inverse correlation between low protein intake and hearing discomfort based on mean hearing thresholds; however, they did not report its association with the degree of hearing loss [75,76]. Protein intake was also significantly associated with increased mean hearing thresholds; however, only the hearing threshold at 1000 Hz exhibited a statistically significant correlation with protein intake. 

A dietary pattern characterized by high protein intake is also associated with a reduced risk of tinnitus (OR of 0.90, 95% CI 0.82 to 0.99) [63]. Similarly, a recent Korean population study reported that insufficient protein intake was associated with an increased risk of tinnitus [77,78,79]. However, little is known about the mechanism through which protein intake affects inner ear function. Nevertheless, numerous studies have shown that inadequate protein intake contributes to muscle atrophy in the elderly population, a phenomenon known as sarcopenia. This illness may cause balance disorders in the elderly and increases the risk of falls. Furthermore, amino acid supplementation was found to increase muscle mass in elderly individuals with impaired glucose tolerance or sarcopenia [80]. 

Although a low-protein diet may impair audio-vestibular function, no current study has specified the amount of protein required in the diet. A high-protein diet is defined as acquiring 20% or more of the total daily calories from proteins. Most high-protein diets are simultaneously high in saturated fat and severely restrict carbohydrate intake [81]. Moreover, high-protein diets may not be appropriate for all individuals, as they may be detrimental to individuals with kidney dysfunction [82].

#### 3.4.3. Low Sugar and Caloric Restriction Diet

People who consume high amounts of carbohydrates, especially simple carbohydrates and added sugars, tend to become obese, which results in other comorbidities, such as cardiovascular diseases, dyslipidemia, and diabetes mellitus [83]. High blood sugar levels can damage small blood vessels and nerves in the inner ear, causing pathological changes in outer hair cells and spiral ganglion cells [84,85]. This results in ischemia and hypoxia in the neural tissues, leading to nerve damage [86].

d-Galactose is a common reducing sugar in the body, and when present at high levels, it can be oxidized to aldehydes and H_2_O_2_ in the presence of galactose oxidase, thereby inducing oxidative stress [87]. Animal-based studies have been utilizing this process to mimic the natural aging process in rats and mice [88].

High carbohydrate and sugar intake are associated with an increased incidence of hearing loss. Glucose-rich and total-carbohydrate-rich diets are predictors of incident hearing loss [61]. High postprandial glycemia may be a potential underlying biological mechanism in age-related hearing loss development [46,76,89,90,91]. Furthermore, a recent study reported that increased consumption of sugary foods was associated with a significantly increased risk of ARHL. Pro-inflammatory foods with high sugar content were also associated with ARHL. Studies assessing the relationship between ARHL and mean blood glucose levels found that the ARHL group had higher, but not significant, mean blood glucose levels [50,92]. Furthermore, a high starch diet is associated with an increased risk of tinnitus [57,93]. 

As previously discussed, several comorbidities, such as obesity and cardiovascular diseases, may accelerate hearing loss by disrupting blood flow in the inner ear [94,95]. Gopinath et al. reported that high glucose and total carbohydrate diets among older adults were associated with an increased risk of developing hearing loss [61,96]. Low-molecular-weight carbohydrate consumption is also correlated with poorer hearing thresholds at high frequencies [58].

In contrast, Spankovich et al. demonstrated that a high carbohydrate intake protects against hearing loss in older adults [97]. This is possible because complex carbohydrates such as cereal fibers, whole grains, and vegetables are associated with a low glycemic index and may lead to earlier satiety and lower energy intake [98]. The protective effects of cereal fibers may override the deleterious effects of other carbohydrate components. Tang et al. reported the potentially protective effects of fiber against tinnitus via vascular risk factors such as cardiovascular disease [99], which may confer a protective effect on hearing by improving insulin sensitivity or reducing postprandial glycemia [100]. Hence, different sources of carbohydrates may have different biological effects on the body and affect hearing capacity differently.

Caloric restriction (CR) extends the lifespan, improves health, and slows ARHL progression in older adults [101]. Furthermore, CR was shown to reduce the incidence of obesity, diabetes, and tumors [102]; protect neurons from neurodegenerative diseases [103]; and decrease inflammation and oxidative damage in aged brains, hearts, and livers in animal models [104]. Several studies have attempted to elucidate whether CR or dietary restriction (DR) can protect against age-related hearing loss in animal models [44,105,106]. DR was shown to maintain the auditory reflex and cellular integrity of the stria vascularis in rats [107]. Seidman et al. utilized a prospective randomized rat model to reveal that the 30%-caloric-restricted group maintained the most acute auditory sensitivity, the lowest quantity of mtDNA deletions, and the least outer hair cell loss [29,108].

#### 3.4.4. Mediterranean Diet

The Mediterranean diet (MedDiet) has attracted considerable attention and has been widely promoted in recent years. It is a plant-based eating plan that includes a daily intake of whole grains, olive oil, fruits, vegetables, beans and other legumes, nuts, herbs, and spices. This dietary pattern contains complex and abundant bioactive compounds, such as several vitamins, minerals, polyphenols, fibers, nitrates, and mono-unsaturated and polyunsaturated fatty acids [109]. MedDiet can reduce the risk of several age-related chronic diseases, such as cardiovascular disease [110], diabetes [111], neurodegenerative diseases [112], and even malignancies [113]. Furthermore, epidemiological studies have shown increased longevity in individuals that adhere to the MedDiet [114,115,116]. In a study conducted in 2019, women who ate healthier diets, including the alternate Mediterranean diet, had lower rates of hearing loss than those who ate less healthy diets [117]. However, no definitive studies have deciphered the exact role of the Mediterranean diet in the pathophysiological mechanisms of ARHL and ARVL. 

The composition of the MedDiet is multivariate and cannot be explained by a single factor. Nevertheless, previous studies have shown that fibers and polyunsaturated fats may be beneficial components of the MedDiet. Given that aging damages cochlear and vestibular function via inflammaging and oxidative stress, the components of the MedDiet carry anti-inflammatory [118] and antioxidant [119,120] effects, which may prevent ARHL and ARVL development. 

In short, a diet rich in fruits, vegetables, and meat but low in fat is associated with reduced odds of hearing difficulties [63].

#### 3.4.5. Different Types of Nutritional Supplements (Vit. A, B, C, D, E, Ca, Mg, Melatonin, and Herbal Remedies)

A wide variety of supplements and herbal remedies are easily accessible to consumers. Certain nutrients and vitamins, such as vitamins A and B (specifically B2, B9, and B12), C, and E, positively influence hearing [76,121,122,123]. A high vitamin B12 intake is also associated with a reduced risk of tinnitus development [63]. 

Folic acid/folate (vitamin B9) plays a crucial role in DNA synthesis and cell proliferation. One study suggested that total folic acid/folate intake is associated with an overall reduced risk of hearing loss in men over the age of 60 [122]. In a randomized clinical trial conducted in the Netherlands, folic acid supplementation improved hearing loss commonly associated with aging; however, it did not affect the decline in hearing high frequencies [121]. A study published in the American Journal of Clinical Nutrition concluded that insufficient vitamin B12 and folate intake may lead to age-related auditory dysfunction [124]. Vitamin B12 may also help treat chronic tinnitus in individuals with vitamin deficiencies [124].

High vitamin D intake is associated with a reduced risk of hearing difficulties. In addition, provitamin A carotenoid and vitamins A, C, and E are known to have antioxidant effects that help prevent and treat hearing loss [124,125]. An animal-based study, wherein dogs were fed a diet rich in antioxidants for the last three years of their lives, showed decreased degeneration at the cochlear base and apex than the control diet group. The antioxidant diet used in this study included vitamin E, l-carnitine, and vitamin C [126]. Furthermore, previous studies also noted that antioxidant-treated subjects had improved auditory sensitivities and fewer mtDNA deletions than control subjects [76,127]. 

Minerals, such as magnesium, zinc, and selenium, are also thought to play a role in the hearing process [123,128]. Magnesium may reduce tinnitus symptoms [129], whereas zinc may reduce tinnitus symptoms only in individuals with zinc deficiency [130,131]. In contrast, high calcium and iron intake was associated with an increased risk of hearing loss [63].

Nevertheless, inconsistent results exist between micronutrients, including vitamins A, B, C, and E, magnesium, and hearing loss. Shargorodsky et al. prospectively evaluated the association between vitamin C, E, B12, and β-carotene intake and the incidence of hearing loss. They found that a high intake of vitamins C, E, B12, or β-carotene does not reduce the risk of hearing loss [122]. Furthermore, vitamin A, C, and E intake is significantly associated with hearing loss [125,132]. Another study reported that there is no statistically significant association between hearing loss and magnesium levels [133]; however, dietary antioxidant intake did not increase the risk of hearing loss. The inconsistency of single-nutrient research may be attributable to differences in study design (e.g., cross-sectional or longitudinal), modalities of hearing measurements (e.g., self-reported or audiometric), and methods for dietary measures (e.g., questionnaires or serum-based). Further large prospective studies are warranted to assess these relationships in older adults. 

Preliminary studies have examined the potential beneficial effects of vinpocetine against memory loss, cancer, and Alzheimer’s disease [134,135]. A clinical study highlighted its effect in treating acquired sensorineural hearing loss and improving hearing [136]. Further, several clinical studies have shown that ginseng reduces tinnitus symptoms and improves the hearing threshold in people with sensorineural hearing loss [137,138].

Coenzyme Q10 is an antioxidant that may prevent and treat certain diseases and improve physical performance, and its supplementation may treat sudden sensorineural hearing loss [139]. 

Ginkgo biloba is the most studied dietary supplement for tinnitus treatment. It is believed to improve tinnitus by increasing inner ear and cerebral blood circulation and protecting against free radicals. However, several clinical trials evaluating its effect on tinnitus have yielded conflicting results [140,141]. 

Tinnitus and sleep are closely related. Melatonin is an essential hormone that regulates the circadian rhythm and protects against free radicals and ototoxic drugs by its antioxidant effects. A review of studies assessing the role of melatonin in tinnitus treatment concluded that it could improve sleep problems caused by tinnitus [142].

Coffee is one of the most common beverages consumed worldwide. Increased coffee consumption had a statistically significant inverse correlation with bilateral hearing loss in the 40–64 years age group, suggesting its protective effect against hearing loss and tinnitus [143]. However, other studies have suggested that caffeine does not strongly affect the peripheral auditory and vestibular systems [144].

Table 1 summarizes the effects of diet on ARHL, ARVL, and related components established from the studies included in this review. So far, there is insufficient evidence to make clinical recommendations regarding the use of vitamins or other supplements for hearing protection, and further studies are needed to determine whether additional intake may influence hearing loss.

### 3.5. The Effects of Lifestyle on the Audio-Vestibular Dysfunction of the Elderly

#### 3.5.1. Exercise

Exercise is beneficial for health as it improves cardiovascular function, physical fitness, and psychosocial health. Regular exercise can also reduce multiple cardiovascular risk factors, such as correcting lipoprotein profiles and lowering fat mass and blood pressure [145]. Sarcopenia is a progressive and generalized skeletal muscle disorder that involves accelerated loss of muscle mass and function and is associated with increased adverse outcomes. It commonly occurs in the older population and has become a growing concern in recent years [146]. Few studies have revealed a potential association between sensory impairments and a greater likelihood of sarcopenia, especially relating to hearing and vision loss. A recent study showed that the prevalence of mild or moderate-to-profound hearing loss was significantly lower in the control group than in the sarcopenia group [147,148]. Higher muscular and performance fitness, such as vigorous activities, are also associated with a lower incidence of hearing loss, particularly high sound frequencies [149]. In contrast, decreased physical function is associated with hearing loss in older adults [150]. A study that enrolled 1180 adults between the ages of 50 and 69 years showed that increased hearing loss was independently associated with a slower gait [151]. However, a valid association between sensory impairment and sarcopenia and its related components could not be established due to the small number of studies assessing this relationship [152]. Furthermore, comprehensive interventional studies on exercise and hearing are lacking. Nevertheless, a growing body of evidence points to the use of exercise as an important therapeutic strategy to prevent and treat sarcopenia. Aerobic exercise ameliorates mitochondria-derived problems, and resistance exercise increases muscle mass and function [147,153,154]. In a CBA/CaJ mouse model, reduced stria vascularis atrophy and capillary loss associated with inflammation were reported in the exercise group, which subsequently delayed ARHL progression [45,155]. 

Dizziness and imbalance are the most common complaints among older adults. Their etiologies are multifactorial; however, peripheral vestibular dysfunction is one of the most commonly reported causes. Age-related vestibular function impairment has also been shown to correlate with the age-related decrease in vestibular hair cell and neuron levels [8,9,155,156,157]. Vestibular rehabilitation can effectively treat unilateral and bilateral vestibular dysfunction and induce faster recovery of posture-locomotor deficits during vestibular compensation, which is associated with a decrease in neurogenesis [158,159]. Animal models have highlighted the positive effects of age and endurance exercise on proprioceptive and vestibular connectivity in motor neurons. Exercise-linked improvement in age-related loss of balance is associated with increased vestibular input to the motor neurons [160]. However, most of the aforementioned studies focused on the benefits of exercise in treating vestibular decline but not on its prevention.

#### 3.5.2. Sleep

Adequate sleep is a subjective issue that varies among individuals. Sleep hygiene is defined as a set of behavioral and environmental recommendations intended to promote healthy sleep. Several sleep pattern changes are associated with aging and include longer times to fall asleep, increased awakenings during the night, and earlier awakenings in the morning. Approximately half of the older population complains of sleep disturbances [161]. A marginal association exists between achieving over eight hours of sleep and impaired high-frequency hearing [162]; however, no evidence of the association between sleep quality and hearing loss has been reported. Nevertheless, several studies have suggested an association between sleep-disordered breathing (SDB) and worse cardiovascular, cognitive, and functional outcomes. It has been hypothesized that sleep disturbances could impair hearing through disturbed energy metabolism and disrupted cochlear blood flow [162,163,164]. Intermittent hypoxia caused by SDB may also cause ischemic injury in the cochlea [163,164,165,166].

#### 3.5.3. Smoking and Alcohol

Smoking may affect the auditory system via the direct ototoxic effects of nicotine and other substances found in cigarette smoke [167] or via vascular effects, such as increased blood viscosity and reduced available oxygen, which can cause cochlear hypoxia [168,169]. Smokers accompanied by exposure to occupational noise or those older than 40 years have an increased prevalence of hearing loss compared to the expected estimate based on the summation of each factor separately [160,161,162,163,164,165,166,167,168,169,170,171]. To determine the risk factors associated with hearing loss in the elderly, 496 subjects with bilateral hearing loss and 2807 age-matched persons without hearing disturbance were recruited, and their lifestyle and medical data were analyzed. Current smokers showed a significantly increased risk of hearing loss compared to non-smokers [172]. Smoking is also associated with peripheral vestibular disorder events, specifically in male participants who smoked for over 30 pack-years [173].

Alcohol consumption has also been linked to hearing loss. One alcoholic drink equivalent contains 14 g of pure alcohol. According to the Dietary Guidelines for Americans, 2020–2025, “drinking in moderation” is defined as a daily alcohol intake of two drinks or less for men and one drink or less for women [174]. Moderate alcohol consumption protects against cardiovascular disease; however, high alcohol consumption is associated with an increased risk [175]. Furthermore, moderate alcohol consumption is inversely correlated with low-frequency hearing loss [176]. A Japanese study revealed that heavy drinkers did not show an increased risk compared with non-drinkers, although a possible selection bias was mentioned [172]. Alcohol intake also interferes with an individual’s balance [177]. A prospective study suggested that alcoholics who were not drinking presented with significant postural imbalance compared to non-alcoholic individuals [178].

#### 3.5.4. Noise Protection

Several studies have established the relationship between noise exposure and age-related hearing loss. A cross-sectional and longitudinal study showed that workplace noise exposure increases the risk of incidental hearing loss in older adults [179]. Fernandez et al. showed that the interaction between noise and aging might require acute synaptopathy to accelerate cochlear aging [180]. Furthermore, a low-intensity noisy environment may delay the onset of age-related hearing loss [181]. ARHL and noise-induced hearing loss share pathophysiological mechanisms, wherein they are both associated with excess free radical formation and cochlear blood flow reduction [182]. Noise exposure is known to induce reactive oxygen species (ROS) generation in the cochlea, and cumulative oxidative stress can be enhanced by relatively hypoxic conditions resulting from impaired homeostasis of cochlear blood supply due to other co-morbidity factors such as atherosclerosis [35]. Noise exposure also affects capillary-perivascular units of the stria vascularis [23,28] and stimulates circulating immunocyte recruitment to the cochlea [22,24,25]. Exposing mice to loud noises led to acute and substantial cochlear afferent terminal loss and slower spiral ganglion cell degeneration. However, no significant cochlear hair cell and pure tone hearing loss was reported, which may have been due to an insufficient exposure period [48,183]. Repeated noise overstimulation over short periods accelerated the time course of hearing loss in ARHL animal models [181].

Noise can also damage the vestibular system [47]. An early study conducted by Mangabeira-Albernaz et al. in 1959 reported that noise damages the peripheral vestibular system [184]. Moreover, morphological studies have also shown cellular damage throughout the peripheral vestibular system, especially in otolith organs [185,186]. Thus, implementing preventive measures to diminish environmental noise exposure is crucial, as they could potentially reduce the burden of ARHL and ARVL. 

Table 2 summarizes the studies analyzing lifestyle effects on ARHL, ARVL, and related components included in this review. The results highlight the importance of maintaining physical activity, fair sleep quality, moderate alcohol consumption, smoking cessation, and avoiding noise exposure.

## 4. Discussions

The inner ear is susceptible to aging effects. Free radical formation, which results from increased oxidative stress, and inflammation typically occur with age and can lead to impaired vascular function, as well as cochlear and vestibular damage. 

Several studies have revealed the harmful effects of a high-fat diet on the inner ear, wherein it induces oxidative stress, mitochondrial damage, and apoptosis. High saturated fat intake was linked to hearing loss, possibly via cardiovascular disease pathways. Obesity and other comorbidities contribute to ARHL by decreasing adiponectin levels. Our previous clinical cohort study highlighted the protective role of adiponectin against ARHL development. On the other hand, a low-saturated fat diet may protect against hearing loss, wherein high polyunsaturated fat intake decreases the incidence of subjective and audiometric hearing loss. 

Insufficient protein intake results in ototoxic side effects, and low protein intake may impair neural function in the auditory system. Some studies have revealed the relationship between protein intake, aural symptoms, and vestibular dysfunction. However, most of the studies were not specific to ARHL and ARVL. No detailed mechanisms have been depicted.

High blood sugar levels can damage small blood vessels and nerves in the inner ear, causing pathological changes in outer hair cells and spiral ganglion cells. High carbohydrate and sugar intake are associated with an increased incidence of hearing loss. Glucose-rich and total-carbohydrate-rich diets are predictors of incident hearing loss. 

The Mediterranean diet has attracted considerable attention and has been widely promoted in recent years. This dietary pattern contains complex and abundant bioactive compounds, such as several vitamins, minerals, polyphenols, fibers, nitrates, and mono-unsaturated and polyunsaturated fatty acids. MedDiet can reduce the risk of several age-related chronic diseases. However, no definitive studies have deciphered the exact role of the Mediterranean diet in the pathophysiological mechanisms of ARHL and ARVL.

A wide variety of supplements and herbal remedies are easily accessible to consumers. Certain nutrients and vitamins positively influence hearing. Nevertheless, inconsistent results exist between micronutrients, including vitamins A, B, C, and E, magnesium, and hearing loss.

Exercise is beneficial for health as it improves cardiovascular function, physical fitness, and psychosocial health. Sarcopenia commonly occurs in the older population and few studies have revealed a potential association between sensory impairments and a greater likelihood of sarcopenia, especially relating to hearing and vision loss. A recent study showed that the prevalence of mild or moderate-to-profound hearing loss was significantly lower in the control group than in the sarcopenia group. Age-related vestibular function impairment has also been shown to correlate with the age-related decrease in vestibular hair cell and neuron levels. Vestibular rehabilitation can effectively treat unilateral and bilateral vestibular dysfunction and induce faster recovery of posture-locomotor deficits. 

Several studies have suggested an association between sleep-disordered breathing (SDB) and worse cardiovascular, cognitive, and functional outcomes. It has been hypothesized that sleep disturbances could impair hearing through disturbed energy metabolism and disrupted cochlear blood flow.

Smoking may affect the auditory system via the direct ototoxic effects of nicotine and other substances found in cigarette smoke. Current smokers showed a significantly increased risk of hearing loss compared to non-smokers. Smoking is also associated with peripheral vestibular disorder events. As for alcohol, moderate alcohol consumption protects against cardiovascular disease; however, high alcohol consumption is associated with an increased risk.

Several studies have established the relationship between noise exposure and age-related hearing loss. It can also damage the vestibular system.

There are several limitations in this review article. Different populations were included in both animal and human studies. Some studies focus on aural symptoms or vestibular symptoms but are not specific to ARHL or ARVL. Some studies enrolled different age groups; not all the studies related to nutrition or lifestyle restricted to the elderly or unified age grouping. The evaluation of nutrition status and lifestyle were extremely complex and multiplex. The exposures and risk factors in each study were limited because most of the exposures and risk factors were mostly evaluated by a questionnaire which was difficult to be quantified or qualified. We did not perform a meta-analysis because during the process, the population, the study design, and the scale of the study were full of heterogeneity which made it unachievable.

## 5. Conclusions

With a continually aging society, the burdens of ARHL and ARVL continue to rise. Thus, preventing rapid auditory and vestibular functional deterioration can greatly improve the quality of life. A fat-rich diet may induce oxidative stress, and low protein intake is associated with hearing discomfort in the elderly. Increased carbohydrate and sugar intake positively correlated with audio-vestibular dysfunction, whereas Mediterranean-style diets were inversely associated with it. Vitamins A, C, and E act as antioxidants that can also minimize the risk of ARHL and ARVL. Maintaining physical activity, fair sleep quality, moderate alcohol consumption, smoking cessation, and avoiding noise exposure are also beneficial. 

However, most studies have focused on ARHL, and little is known about ARVL. Furthermore, previous studies comprised mainly observational or interventional animal models. Therefore, additional large-scale studies are required for further verification, as human longitudinal studies are essential to understand individuals’ risk of diseases much earlier in life and to inform health choices and medical care options.

## Figures and Tables

**Figure 1 nutrients-14-04720-f001:**
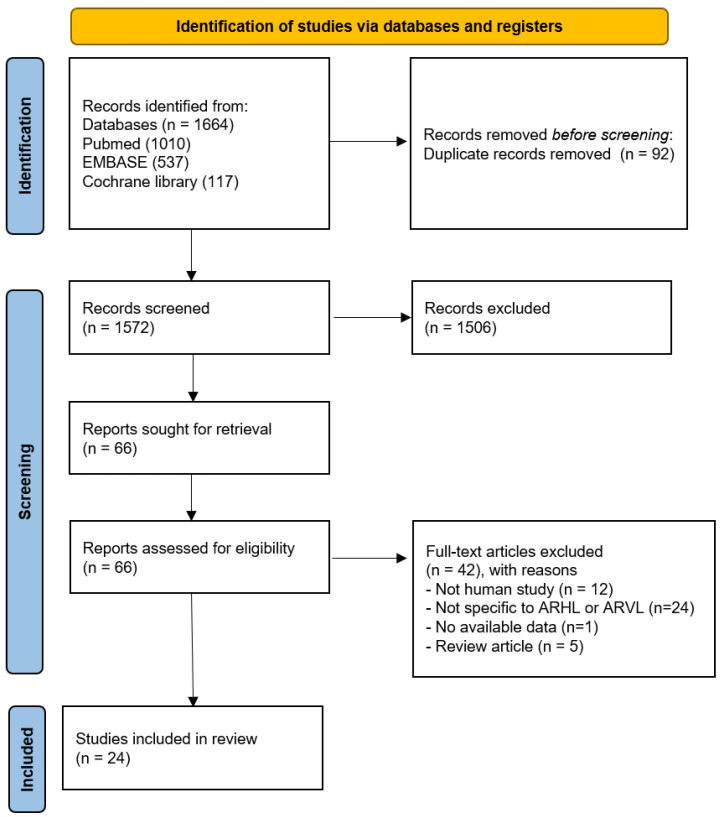
Flow diagram of literature research.

**Figure 2 nutrients-14-04720-f002:**
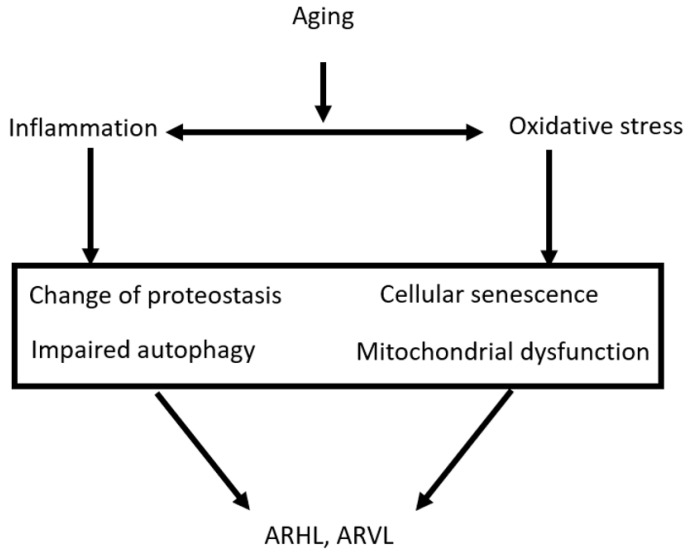
Pathogenetic mechanisms linked to ARHL and ARVL Summary of the pathogenetic mechanisms of age related hearing loss and vestibular loss. Aging contributes to inflammation and accumulation of oxidative stress which result in change of proteostasis, cellular senescence, impaired autophagy, and mitochondrial dysfunction. Ultimately, the inner ear is damaged by the detrimental substances.

**Table 1 nutrients-14-04720-t001:** The effects of diets on ARHL, ARVL, and related components.

	Study Included	Age (Mean)	N	Male	Female	Category	Test	Odds Ratio	95% CI	Relative Risk	95% CI	*p*-Value	Outcomes
Fat	Gopinath/2010 [72]	>50	2442	1053	1389	Cross sectional	PTA, questionnaire	0.76	0.60–0.97			NA	Inverse association between higher intakes of long-chain n-3 PUFAs and HL.
	Curhan/2014 [73]	NA	65,215	0	65,215	Prospective cohort	Self-reported hearing loss, questionnaire			0.85	0.80–0.91	<0.001	Higher intake of long-chain omega-3 PUFAs are associated with lower risk of HL in women.
	Rosenhall/2015 [58]	70–75	524	249	275	Cross sectional	PTA, dietary history	NA	NA			<0.05	Good hearing and a high consumption of fish in the male group.
	Kim/2015 [75]	68.3	4615	2049	2566	Cross sectional	PTA, nutritional survey	0.82	0.71–0.96			0.011	Low fat intakes are associated with hearing discomfort.
	Dawes/2020 [63]	40–69 (55.8)	34,576	15,974	18,602	Cross sectional	Self-reported hearing problems, questionnaire	1.16	1.08–1.24			<0.05	Substantial impact of diet on levels of tinnitus and hearing difficulties.
Cholesterol	Gopinath/2011 [52]	>67	2447	1053	1394	Cross sectional and cohort	PTA, questionnaire	1.34	1.00–1.80			0.04	High cholesterol diet could have adverse influences on hearing.
Protein	Kim/2015 [75]	68.3	4615	2049	2566	Cross sectional	PTA, nutritional survey	0.81	0.67–0.96			0.017	Low protein intakes are associated with hearing discomfort.
Carbohydrates	Gopinath/2010 [61]	>65	2448	NA	NA	Cross sectional	PTA, questionnaire	1.77	1.04–3.00			0.03	Higher intake of total carbohydrate was a predictor of incident HL.
	Rosenhall/2015 [58]	70–75	524	249	275	Cross sectional	PTA, dietary history	NA	NA			<0.05	Poor high frequency hearing and a high consumption of food rich in low molecular carbohydrates in both genders.
	Sardone/2020 [50]	>65	734	425	309	Cross sectional	PTA, questionnaire	NA	NA			0.05	High-sugar content food is associated with positive ARHL-status.
	Lampignano/2021 [92]	>65	734	403	331	Cohort	PTA, questionnaire	0.998	0.996–0.999			NA	Lesser carbohydrate intake is associated with age-related central auditory processing disorder.
	Tang/2021 [99]	>50	1730	NA	NA	Longitudinal cohort	Questionnaire	1.54	1.07–2.22			NA	Modest associations between intake of dietary fiber and incident tinnitus.
Mediterranean diet	Dawes/2020 [63]	40–69 (55.8)	34,576	15,974	18,602	Cross sectional	Self-reported hearing problems, questionnaire	0.89	0.83–0.96			0.024	Dietary patterns high in fruit and vegetables and meat and low in fat was associated with reduced odds of hearing difficulties.
Vitamins	Durga/2007 [121]	60	728	522	206	Randomized controlled trial	Audiometry	NA	NA			0.02	Folic acid supplementation slowed the decline in hearing of the speech frequencies.
	Gopinath/2011 [132]	>50	2956	NA	NA	Cross-sectional and 5-year longitudinal analyses	PTA, questionnaire	0.53	0.30–0.92			0.04	Dietary vitamin A intake was significantly associated with the prevalence of HL.
	Gopinath/2011 [132]	>50	2956	NA	NA	Cross-sectional and 5-year longitudinal analyses	PTA, questionnaire	0.86	0.78–0.98			NA	Dietary vitamin E intake was significantly associated with the prevalence of HL.
	Kang/2014 [123]	50–80 (62.53)	1910	810	1100	Cross sectional	PTA, questionnaire	−0.012	−0.022–0.002			<0.05	Dietary intake of vitamin C was associated with better hearing in the older population.
Minerals & others	Lee/2018 [143]	>65	2184	NA	NA	Cross sectional	PTA, questionnaire	0.76	0.56–1.03			0.0778	No significant decreases in bilateral HL were observed in the >65 years age groups.
	Dawes/2020 [63]	40–69 (55.8)	34,576	15,974	18,602	Cross sectional	Self-reported hearing problems, questionnaire	1.2	1.08–1.34			0.02	Higher intakes of calcium were associated with increased odds of tinnitus.
	Dawes/2020 [63]	40–69 (55.8)	34,576	15,974	18,602	Cross sectional	Self-reported hearing problems, questionnaire	1.2	1.05–1.37			0.007	Higher intakes of iron were associated with increased odds of tinnitus.

PTA: pure-tone audiometry; HL: hearing loss; NA: not applicable.

**Table 2 nutrients-14-04720-t002:** The effects of lifestyles on ARHL, ARVL and related components.

	Study Included	Age	N	Male	Female	Category	Test	Odds Ratio	95% CI	*p*-Value	Outcomes
Exercise	Kawakami/2021 [149]	43–54	2765	1767	998	Prospective cohort	Muscular and performance fitness index, PTA	0.79	0.71–0.88	<0.001	Higher muscular and performance fitness is associated with a lower incidence of HL.
Sleep deprivation	Martines/2016 [164]	38–55	160	103	57	Cross sectional	Polysomnography, PTA, TEOAE	NA	NA	<0.05	A more marked high-frequency hearing loss in case of severe OSAS.
	Ekin/2016 [165]	20–60	66	40	26	Cross sectional	Polysomnography, PTA	NA	NA	<0.001	Snoring may cause hearing loss at extended high frequencies.
	Jiang/2021 [162]	>70	632	325	307	Cross sectional	PTA, questionnaire	NA	–0.34–5.24	>0.05	Longer sleep duration is marginally associated with poorer high-frequency hearing among older adults sleeping >8 hours.
Smoking and alcohol	Itoh/2001 [172]	>60	496	454	42	Cross sectional	PTA, questionnaire	2.1	1.53–2.89	<0.001	Current smokers had a significantly increased risk of HL.
	Itoh/2001 [172]	>60	496	454	42	Cross sectional	PTA, questionnaire	0.96	0.57–1.64	0.021	Heavy drinkers showed no increased risk of HL.
	Ferrite & Santana/2005 [171]	20–55	535	535	0	Cross sectional	PTA, questionnaire	7.65	4.43–13.23	NA	Synergistic effect of smoking, noise exposure and age on HL.
	Pouryaghoub/2007 [170]	24–67	206	206	0	Cross sectional	PTA	7.4	4.1–13.4	<0.001	Smoking can accelerate noise induced HL.
	Gopinath/2010 [176]	Mean 66.6	2815	1218	1597	Cross sectional	PTA, questionnaire	0.75	0.57–0.98	0.04	A protective association between the moderate consumption of alcohol and hearing in older adults.
	Wada/2017 [173]	mean 65.3	393	133	260	Retrospective cohort	Medical record	2.7	1.32–5.53	0.006	Smoking history of >30 pack-years increased the risk of new onset peripheral vestibular disorder.
Noise	Gopinath/2021 [179]	>50	1932	NA	NA	Cross sectional	PTA, questionnaire	1.39	1.13–1.71	NA	Workplace noise exposure increased the risk of incident hearing loss in older adults.

PTA: pure-tone audiometry; TEOAE: transient evoked otoacoustic emissions; HL: hearing loss; NA: not applicable.

## Data Availability

Not applicable.

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
