# Peer review of "Effects of Diet and Lifestyle on Audio-Vestibular Dysfunction in the Elderly: A Literature Review"

_nutrients, 2022, doi:10.3390/nu14224720_

Round 1
Reviewer 1 Report
The authors of the paper Effects of Diet and Lifestyle on Audio-vestibular Dysfunction 2 in the Elderly: A Comprehensive Review
reports a much-needed perspective on age-related disorders. Due to the need for high-quality papers in this field, I urge the authors to consider the following suggestions that, in my opinion, would greatly improve the context and conclusion.
Major comments:
The method section is very scarce in information.
- I suggest to rethink your search strategy and include other medical/healthcare databases such as Embase, CDRS, and others. This will not only increase the number of studies but also contribute to the generalizability of the findings.
- Is there a Prospero-published protocol for the search strategy? If so, please state the number.
- Are there any pre-defined articles that justify the incentive for the current work?
- Clarify and report your inclusion/exclusion criteria of the papers( languages, human samples only or not, were book chapters, conference material, or case reports included?)
- Study design - please summarize and clarify the number of papers that you found with risk factors that passed the inclusion criteria. The conclusions of a cross-sectional study and a cohort study are very different and this can improve the comparability of results.
- Is the study population consistent among studies?
- How were the cases of ARHL defined? Conversely, how were the exposures and risk factors limited?
- What are the reported measures of association in the papers included in the review? Again, OR s and RR have different conclusions.
- How was the screening done? Was it a blinded screening with multiple investigators involved?
- What is the quality assessment that you performed to ensure the quality of the study was sufficient for further consideration? Please consider using Newcastle Ottawa Scale for quality control.
- Is there a reason why a meta-analysis was not performed? If, so please state it clearly in the manuscript.
Result section
- Please start this section by describing the papers and summarizing the findings into the subsequent categories.
- In section 3.1.2. Other factors: genetics, environment, and medication, please consider findings from a recent GWAS on ARHL and comment on the genes expressed in the stira vascularis that contribute to hearing loss due to oxidative stress and vascularization of the cochlea. Since you already mention the etiology of ARHL, it read incomplete without commenting on the most recent findings.
- Please clarify the number of papers you identified for each category and the study design of those papers
- Please summarize in a Table the measured association of each study for each risk factor (Ors and RRs with CI95% with p-values)
Section 3.2.
- This is the critical and essential summary of the tremendous amount of work the authors did.
- I strongly advise clarifying the categories you have mentioned. For example, low fat and low cholesterol diet - were that randomized trial studies, questionnaire data, were people followed up, and how many papers investigated the effect of a low-fat diet on ARHL? More importantly, what food groups were included, was it animal fat or plant-based? What was the control group? What of the food groups were significantly associated with ARHL and ARVL?
- The previous comment should be considered for all macro and micronutrients.
Minor comments:
- A figure that can summarize the findings will elevate the quality of the paper significantly.
- Forest plots are important to do in order to inspect the effect of the studies involved
Conclusion: This is immensely important work in the research of age-related disorders since most of the chronic and aging disorders are preventable by modifiable factors that have been neglected in research. Despite the ambiguity of nutritional results, I believe there are ways to improve current standards of conducting study designs involving nutritional science. The work of the authors of this paper could be a baseline upon which this field can grow.
Reviewer 2 Report
Comments and Suggestions for Authors
The manuscript provides a comprehensive and carefully analyzed review of the literature related to the mentioned topic. It has a logical structure and very well describes diet and lifestyle's effects on audio-vestibular dysfunction.
However, specific corrections are needed to improve the manuscript, so the review needs revision. Please see below.
1. Method:
a. Add a statement on the inclusion and exclusion criteria of the studies included in the review.
b. How did you define literature search before the beginning of the investigation? Are there any criteria for language restrictions, publication types, study methods, or years to search? Describe it and describe efforts to decrease selection bias.
c. Describe if you did a secondary review of bibliographies of original review papers.
d. How did you rank (weigh) the quality of the studies included in the literature review? If you didn't, mention it in the Limitations section.
2. Results:
a. Provide a table or chart of the process of study selection.
b. In Table 1 and Table 2, use a different title for the column instead of "Note," which better describes the outcomes you have listed.
c. This section should provide an objective report of the findings, so the authors should present interpretation in the Discussion section of the manuscript.
3. Include the "Discussion" section. Just copy-paste text from the "Results."
4. Provide detailed limitations and strengths you observed with the literature review. You can add it at the end of the "Discussion" section.
Specific comments:
In line 76, correct the abbreviations AVRL (it should be ARVL)
In line 120, correct the order of enumeration. Write 3.1.3. instead 3.1.2.
In line 285, the abbreviation "e.g." seems redundant. Delete it.
In line 464, the sentence "Thus, preventing rapid functional deterioration can greatly improve the quality of life" seems insufficiently precise. "Functional deterioration" of what? Define it more precisely.
In the Reference section:
In lines 482 and 796, correct the way of citing references. According to “Instructions for Authors“ by Nutrients, when you refer to sites, references should be described as follows: Title of Site, Available online: URL (accessed on Day Month Year).
Round 2
Reviewer 1 Report
Thank you to the authors for considering my comments and for their clear attempt to improve the article.
Reviewer 2 Report
The authors have taken into account and adequately responded to all comments. The quality of the article is now at a much higher level, but the authors may still consider that the study's results could be nuanced more subtly.
I commend the hard work in revising this manuscript.